# A case-control regression analysis of liver enzymes in obesity-induced metabolic disorders in South Asian females

**Tamseela Mumtaz**[1]☉*, **Kainat Tariq**[1]☉, **Khadija Kanwal**[2], **Zainab Tariq**[1]

**1** Department of Zoology, Government College Women University Faisalabad, Punjab, Pakistan,
**2** Department of Statistics, Government College Women University Faisalabad, Punjab, Pakistan

☉ These authors contributed equally to this work.
* dr.tamseelamumtaz@gcwuf.edu.pk

**Data Availability Statement:** All relevant data are within the manuscript and its Supporting Information files.

**Funding:** The author(s) received no specific funding for this work.

## Abstract

Excessive body weight may disrupt hepatic enzymes that may be aggravated by obesity-related comorbidities. The current case-control study was designed to evaluate the extent of liver enzyme alteration in obesity-related metabolic disorders. Obese females with BMI ≥ 30 suffering from metabolic disorders were grouped according to existing co-morbidity and their hepatic enzymes were compared with non-obese healthy females. The resultant data was subjected to analysis of variance and mean difference in liver enzymes were calculated at P = 0.05. Analysis of variance indicated that obese diabetic and obese hypertensive females had almost 96% and 67% increase in the concentration of gamma-glutamyl trans-ferase than control, respectively (P<0.0001). The obese females suffering from diabetes and hypertension exhibited nearly 54% enhancement in alanine transaminase level (P<0.0001) and a 17% increase in aspartate aminotransferase concentration (P = 0.0028). Obesity along with infertility decline liver enzyme production and a 31% significant decline in aspartate aminotransferase was observed while other enzyme concentrations were not significantly altered. Regression analysis was performed on the resultant data to understand the association between liver enzyme alteration and the development of metabolic diseases. Regression analysis indicated that obese diabetic and obese diabetic hypertensive women had 20% production of normal liver enzymes and 80% enzymes produced abnormally. Obese hypertensive and obese infertile females had only 5% and 6% normal production of liver enzymes, respectively. This research leads to the conclusion that the ability of the liver to function normally is reduced in obesity-related diabetes and hypertension. This may be due to inflamed and injured liver and poses a serious threat to developing fatty liver disease and ultimately liver cirrhosis.

## Introduction

Metabolic syndrome (MetS) is a collection of cardiometabolic perils including abdominal obesity and overweightness, increased blood glucose levels and hypertension [1]. It is identified

**Competing interests:** The authors have declared that no competing interests exist.

**Abbreviations:** ALT, Alanine Aminotransferase; ANOVA, Analysis of variance; AST, Aspartate Aminotransferase; BMI, Body Mass Index; CVD, Cardiovascular Disease; DBP, Diastolic Blood Pressure; ERC, Ethical Review Committee; GGT, Gamma-Glutamyl Transferase; HC, Hip Circumference; HDL, High Density Lipoproteins; IFCC, International Federation of Clinical Chemistry; IR, Insulin Resistance; MetS, Metabolic Syndrome; NAFLD, Non-Alcoholic Fatty Liver Disease; PCOS, Polycystic Ovary Syndrome; RBS, Random Blood Sugar; rpm, Revolutions per minute; SBP, Systolic Blood Pressure; SEM, Standard Error Mean; T2DM, Type 2 Diabetes Mellitus; WHR, Waist to Hip Ratio; WC, Waist Circumference.

when three out of the following five conditions are present: increased waist circumference, high fasting glucose levels, high blood pressure, raised triglycerides and decreased high-density lipoproteins (HDL) [2]. Metabolic syndrome distresses approximately 30–40% of individuals till they reach 65, caused mainly by weight gain and fat accumulation within abdomen due to genetic or non-genetic factors [3].

Fatty liver and raised hepatic enzyme levels are frequently observed in obesity, serving as trademarks of non-alcoholic fatty liver disease (NAFLD), These enzymes are also raised in metabolic syndrome and insulin resistance (IR) [4]. Insulin resistance is the main factor associated with MetS and NAFLD which has been linked to accumulation of unnecessary fat in areas such as liver, promoting inflammation and stressing the endoplasmic reticulum. This stress and inflammation set up and exacerbate IR [5]. Increased body fat, raised BMI, high insulin level and insulin resistance elevate the levels of liver enzymes; alanine aminotransferase (ALT), aspartate aminotransferase (AST), and gamma-glutamyl transferase (GGT) [6–8]. Elevation of ALT and AST also occurs when there is hypertension and diabetes, which have adverse effects on the heart leading to coronary heart disease (CHD) [9]. Concentrations of GGT even in normal ranges are associated with hypertension in subjects with abdominal obesity, suggesting a role of fatty liver in obesity related hypertension [10] while on the other hand, elevated concentrations of GGT are associated with the increased probability of developing diabetes [11]. Previous studies showed greater circulating levels of hepatic enzymes in individuals with metabolic syndrome and IR and revealed a significant association of these liver enzymes with the likelihood of developing T2DM [12]. Obesity along with reduced liver functioning upsurges the risk of diabetes. By decreasing body weight, liver function can be better and the risk of diabetes development can be lower [13,14].

Considering the above literature, it is evident that the alteration of liver enzymes is associated with obesity. The current study, therefore, aimed to evaluate the correlation of liver enzymes with the obesity-related metabolic syndrome suffering from diabetes, hypertension, diabetes+hypertension, and infertility. The research also seeks to determine how much liver enzyme alterations may contribute to the onset of metabolic syndrome.

## Materials and methods

### Study design

A case-control study was designed to evaluate obesity-induced metabolic diseases and their association with liver enzymes. Obesity-induced diseases included in the study were hypertension, diabetes and infertility. The study design was placed before the Ethical Review Committee (ERC) of Government College Women University and got approved vide letter # GCWUF/IERC/21/132 dated 22-02-2021. The study started in August 2021 and ended in March 2023.

### Study population and sample size

For this study, 162 females with BMI of ≥25 were interrogated. Among them, overweight and obese females were segregated for initial scrutiny and the women who were either overweight or obese without suffering from any comorbidity were excluded at this stage. The females who had intentionally lost their weight during the study to reduce the impact of disease were also excluded. All obese females with BMI≥30 (n = 120) who were suffering from diabetes, hypertension or infertility due to obesity were categorized into four subgroups based on obesity-related co-morbidities. Obese females suffering from diabetes were clustered as Group 1, whereas obese females with hypertension were categorized into Group 2. The obese women suffering from hypertension and diabetes were classified as Group 3. The obese infertile females were clustered into Group 4. All the groups had an equal number of subjects (n = 30).

For comparison, 120 females with BMI≤25 and having no health issues were selected as control (n = 120). As this was a case-control study, the ratio of case-to-control was set as 4:1 which means 4 healthy females were recruited against 1 morbid obese female. This sample size was calculated by G* Power software by applying F family statistical test, "The ANOVA: Fixed effects, omnibus, one-way". The applied analysis was post hoc: Compute achieved power. The input parameter included: Effect size f = 0.40, α err prob = 0.05. The output power of analysis (1-β err prob) was 0.9996582. The study plan was discussed with the participants, and it was made sure that all subjects participated voluntarily without any incentive. A written informed consent was taken and signed by the participants. A comprehensive questionnaire was designed to collect personal information, disease and demographic history. The personal information was kept confidential and each individual was assigned an anonymous identification number to maintain confidentiality (S1 File).

**Inclusion and exclusion criteria.** The age of the participants ranged from 23–60 years and special care was taken to ensure that no participant was over sixty years because this is the part of the age where the risk of getting diseases is high. It was made sure that all the selected females had BMI ≥30 kg/m$^2$ and had developed comorbidity because of obesity. They must have one of the obesity-induced comorbidities such as diabetes, hypertension, and infertility. They should not suffer from any viral disease or any chronic illness like hepatitis, respiratory or gastrointestinal diseases, drug or alcohol usage, or hyperthyroidism that might lead to obesity. It was also made sure that females who were treated with antihypertensive or antidiabetic drugs that may affect liver function should not be part of the study. Women with less than a 30 BMI were not fit for participation in the study and were excluded. Those obese women who did not have any obesity-related diseasealso not considered (S2 File).

## Blood sample collection and processing

Intravenous blood was collected from selected females through venipuncture and poured in 3 ml EDTA-coated vacutainers to prevent agglutination of blood. Blood was centrifuged at 3000 rpm for 15 minutes to separate the plasma. All personal and laboratory safety measures were ensured while sampling. Liver enzymes: ALT, AST, and GGT were determined from plasma using commercially available kits (ALT cat # GL732AL, AST cat # GL733AS and GGT cat # GL705GT) made by Bioactiva diagnostica GmbH with the help of a fully automated biochemistry analyzer (FA-200 Clindiag) and clinical chemistry analyzer (Erba Mannheim CHEM-7, Germany). All standard protocols were recommended by the International Federation of Clinical Chemistry (IFCC).

## Statistical analysis

Obtained data were subjected to statistical analysis through ordinary one-way analysis of variance (Ordinary one-way ANOVA) using Graph-Pad Prism (V.6.0) and difference in mean values was observed at P = 0.05 by performing Dunnett's multiple comparison test, with a single pooled variance. Further, regression analysis was performed through SPSS (V.21) to establish the relationships between liver enzyme variation and the metabolic ailments under study.

## Results

### One-way analysis of variance of anthropometric measurements in obese females

The waist-to-hip ratio was higher in obese diabetic and obese diabetic hypertensive women (1.00 ±0.01 and 1.02±0.01, respectively). Obese infertile females had the least rise in WHR

**Table 1. Anthropometric and biophysical measurements of participants in control and obese groups with metabolic disorders.** Values are mean±SEM.

| Variables | Groups | | | | | |
|---|---|---|---|---|---|---|
| | Control | Ob DM | Ob HTN | Ob DM+HTN | Ob Infertile | P-value |
| WC (inches) | 28.90±0.41 | 43.40±0.83 | 44.55±0.86 | 46.50±0.55 | 40.03±0.88 | <0.0001 |
| HC (inches) | 34.50±0.45 | 43.67±0.78 | 46.14±0.82 | 45.63±0.74 | 43.10±0.81 | <0.0001 |
| WHR | 0.84±0.01 | 1.00±0.01 | 0.97±0.01 | 1.02±0.01 | 0.93±0.01 | <0.0001 |
| BMI (kg/m$^2$) | 21.94±0.37 | 32.44±0.98 | 33.26±1.20 | 34.60±0.96 | 30.06±0.98 | <0.0001 |
| RBG (mg/dL) | 104.5±1.69 | 232.6±13.30 | 105.3±2.83 | 239.1±13.50 | 100.3±2.03 | <0.0001 |
| SBP (mmHg) | 113.7±1.31 | 124.0±1.76 | 147.7±3.02 | 144.7±3.41 | 117.7±1.90 | <0.0001 |
| DBP (mmHg) | 74.33±1.24 | 81.33±0.92 | 92.67±1.51 | 90.67±1.51 | 75.33±1.33 | <0.0001 |
| Cholesterol (mg/dL) | 143.5±1.36 | 176.3±4.55 | 171.3±5.712 | 186.8±7.295 | 141.3±3.387 | <0.0001 |

Significant at P<0.05 at 95% CI; Ob: Obese, DM diabetes mellitus, HTN: Hypertension, BMI: Body mass index; RBG; random blood glucose; SBP: Systolic blood pressure; DBP: Diastolic blood pressure; WC: Waist circumference; HC: Hip circumference; WHR: Waist-to-hip ratio.

(0.93±0.01) as compared to normal females (0.84±0.01). On the other hand, all the women in the study were moderately obese having a BMI of 30 < 34.9 (obese class 1). The BMI of females with metabolic disorders tremendously increased and the mean BMI of obese diabetic females reached up to 32.44±0.98. A similar trend of obesity was noticed in obese hypertensive females with a mean value of 33.26±1.20. When hypertension and diabetes co-existed, the body gained extra mass, and mean BMI increased by up to 36.6% than healthy females. Obese infertile females, however, had an average BMI of 30.07±0.80.

The random blood sugar (RBS) was significantly higher in the diabetic and diabetic hypertensive groups (P<0.0001). Obese hypertensive women also had a minor increase in their RBS level but the increase was within the normal limit (105.3±2.83). Contrary to this, the obese infertile females had a slight decrease in their RBS levels and again, the decrease was within the normal range (100.3±2.03). The systolic and diastolic blood pressure (SBP and DBP) were found to be significantly higher in the hypertensive and diabetic hypertensive groups as expected. Still, the diabetic group also showed an increase in SBP (124.0±1.76) implying stress on the heart by diabetes in association with the progression of obesity. Infertile obese females had normal blood pressure.

Surprisingly cholesterol level was found higher in obese diabetic women (176.3±4.55) than the obese hypertensive females (171.3±5.712). When both diseases were combined, the level shot up to 186.8±7.29 which was 30% higher than the healthy women's cholesterol level (Tables 1 and 2).

**Table 2. Percentage variation in biophysical measurements of healthy control and morbidly obese groups.**

| Groups | Variables | | | | | | | | |
|---|---|---|---|---|---|---|---|---|---|
| | WC% | HC% | WHR% | BMI% | Pulse rate % | RBG% | SBP% | DBP% | Cholesterol % |
| Control Vs Ob. DM | 50.1****↑ | 26.6****↑ | 18.3↑ | 47.9****↑ | 5.6**↑ | 122.6****↑ | 9.0*↑ | 9.3**↑ | 22.7****↑ |
| Control Vs Ob. HTN | 54.1****↑ | 33.7****↑ | 14.9↑ | 51.7****↑ | 12.9****↑ | 0.8↑ | 29.8****↑ | 24.5****↑ | 19.2***↑ |
| Control Vs Ob. DM+ HTN | 60.9****↑ | 32.3****↑ | 21.3↑ | 57.8****↑ | 10.9****↑ | 128.8****↑ | 27.1****↑ | 21.8****↑ | 30.0****↑ |
| Control Vs Ob. infertile | 38.5****↑ | 24.9****↑ | 10.3*↑ | 37.1****↑ | 0.2↓ | 4.0↓ | 3.4↓ | 1.2↓ | 1.7↓ |

↑ indicate an increase in value as compared to control.

\* significant at P = 0.05

\*\* significant at P = 0.01

\*\*\* significant at P = 0.001

\*\*\*\* significant at P = 0.0001.

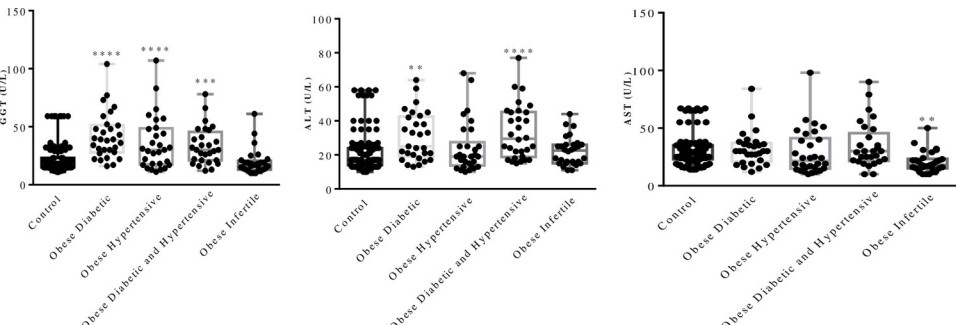

**Fig 1. Individual GGT (U/L), ALT (U/L), and AST (U/L) concentrations in control (n = 120) and obese females with allied metabolic disorders (n = 30 in each metabolic disorder).** The figure illustrates the graphical presentation of the three liver enzymes in each subject concerning the control.

**One-way analysis of variance of biochemical measurements in obese females.** Levels of liver enzymes in obese females suffering from allied diseases are greatly altered. Among the three liver enzymes studied, GGT was found to be significantly higher in obese diabetic and obese hypertensive females (P<0.0001), but its increase was non-significant in infertile obese females. Alanine aminotransferase also showed significant enhancement in obese diabetic and hypertensive females (P<0.0001) while, AST, on the other hand, showed negligible elevation in obese diabetic and obese diabetic hypertensive females. All hepatic enzymes except AST in infertile obese females remained within the normal range (Fig 1).

## The relationship of obesity with cholesterol is a key health indicator in women (Fig 2A–2E)

The correlation between BMI and cholesterol levels in the obese diabetic group revealed an r-value of -0.07122 which indicates a non-significant very small negative relationship between X and Y, p = 0.654 (Fig 2A).

## Production of liver enzymes altered in morbidly obese females

Liver function tests of obese diabetic females according to the regression line are given in Table 3.

In this data model, we take more interest in R Square which indicates how much variation covers, and we see it is only 20% of enzymes are made by the liver properly and 80% are deranged due to obese diabetes.

Liver function tests of obese hypertensive females according to the regression line are given in Table 4 (Fig 3B).

When we look at R Square in this data model, which covered more variations, we found that only 5% of liver enzymes were produced correctly and 95% of enzyme production is abnormal due to obesity-induced hypertension. Liver function tests of obese diabetic hypertensive females according to the regression line are given in Table 5 (Fig 3C).

According to this data model's R Square, (which shows how much variation is covered), only 20% of the enzymes produced by the liver are produced appropriately, while the remaining 80% are out of whack due to obesity-induced hypertension and diabetes. Liver function tests of obese infertile females according to the regression line are given in Table 6 (Fig 3D).

According to this data model only 6% of enzymes are appropriately produced by the liver and 94% of production is deranged due to obesity-related infertility.

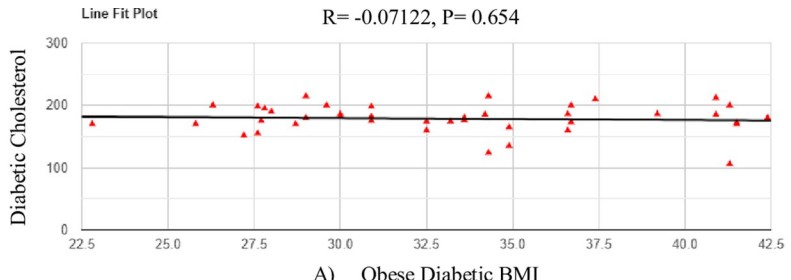

A) Obese Diabetic BMI

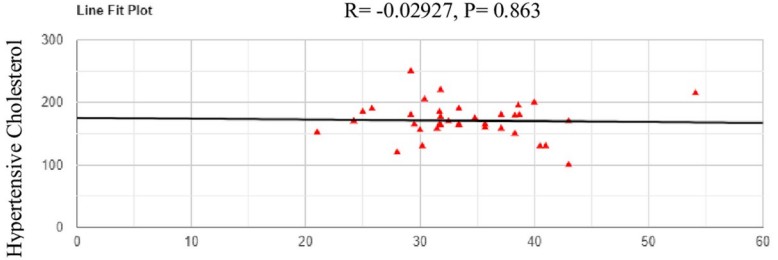

B) Obese Hypertensive BMI

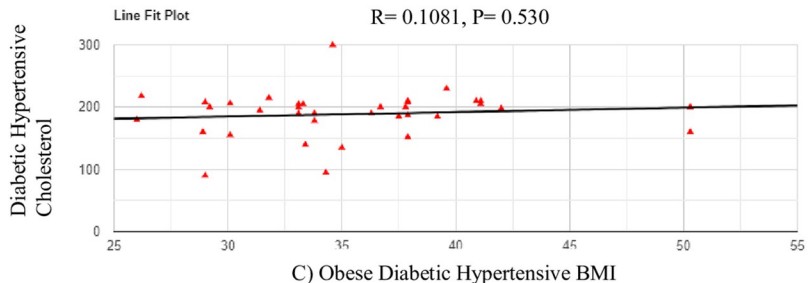

C) Obese Diabetic Hypertensive BMI

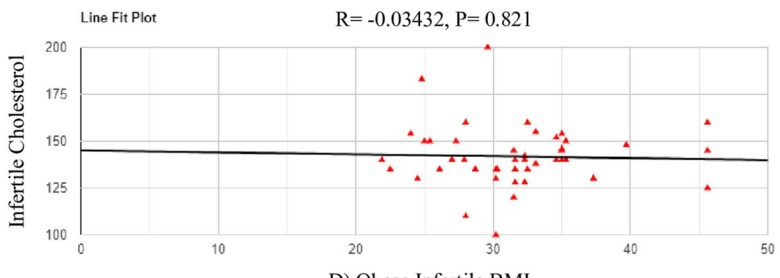

D) Obese Infertile BMI

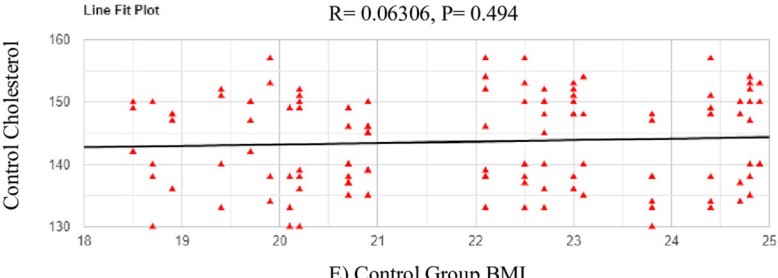

E) Control Group BMI

**Fig 2. Line fit scatter plot for obese diabetic BMI vs cholesterol levels. A.** The x-axis represents the BMI whereas the Y-axis represents the cholesterol values. The correlation between BMI and cholesterol levels in the obese hypertensive group revealed an r-value of -0.02927 which indicates a non-significant very small negative relationship between X and Y, p = 0.863 (Fig 2B). **B. Line fit scatter plot for obese hypertensive BMI vs cholesterol levels**. The x-axis represents the BMI whereas the Y-axis represents the cholesterol values.The correlation between BMI and cholesterol levels in the obese diabetic hypertensive group revealed an r-value of 0.1081 which indicates a non-significant small positive relationship between X and Y, p = 0.530 (Fig 2C). **C. Line fit scatter plot for obese diabetic hypertensive BMI vs cholesterol levels**. The x-axis represents the BMI whereas the Y-axis represents the cholesterol values. The correlation between BMI and cholesterol levels in the obese infertile group revealed an r-value of -0.03432 which indicates a non-significant very small negative relationship between X and Y, p = 0.821 (Fig 2D). **D. Line fit scatter plot for obese infertile BMI vs cholesterol levels**. The x-axis represents the BMI whereas the Y-axis represents the cholesterol values. The correlation between BMI and cholesterol levels in the control group revealed an r-value of 0.06306 which indicates a non-significant very small positive relationship between X and Y, p = 0.494 (Fig 2E). **E. Line fit scatter plot for control BMI vs cholesterol levels**. The x-axis represents the BMI whereas the Y-axis represents the cholesterol values.

Liver function of obese healthy control females according to the regression line and R -Square are given in Table 7 (Fig 3E).

In this data model, we take more interest in R Square which indicates 20% variation covered and 80% enzymes made by the liver properly.

## Liver enzymes are not equally altered in all obesity-induced metabolic disorders

According to the data sample in Table 8, we conclude that we cannot reject the $H_1$ hypothesis (alternate hypothesis) and we are not able to accept the $H_o$ hypothesis (null hypothesis) which states that all enzymes work equally based on the value of alpha $\alpha = 0.05$. In obese diabetic, hypertensive, diabetic hypertensive, and infertile groups the observed p-values are greater than 0.05, indicating that all enzymes are not working equally in comorbid groups. On the contrary, the p-value in the control group is less than 0.05 implying that all enzymes are working equally in the control group.

Further, if alpha changes or the sample data variates, results may vary.

ANOVA seeks to determine the difference in mean at each level of a factor. According to the results below, the results are not statistically significant in the comorbid groups. The whole factors are disturbed strongly and affect the liver enzymes. On the contrary, in the control group, the whole factors are not disturbed and do not affect the liver.

## Discussion

The liver is a vital organ where the metabolism of glucose takes place along with its uptake, synthesis, and storage. Enhanced activities of liver enzymes; ALT, AST, and GGT serve as

**Table 3. Regression model for liver function tests of obese diabetic females.**

| Coefficients[a] | | | | | | | | |
|---|---|---|---|---|---|---|---|---|
| Model | | Unstandardized Coefficients | | Standardized Coefficients | T | Sig. | 95% Confidence Interval for B | |
| | | B | Std. Error | Beta | | | Lower Bound | Upper Bound |
| 1 | (Constant) | 11.068 | 5.407 | | 2.047 | 0.051 | 24.96 | 38.39 |
| | ALT | 0.145 | 0.127 | 0.291 | 1.139 | 0.265 | 25.47 | 43.05 |
| | AST | 0.059 | 0.092 | 0.157 | 0.639 | 0.528 | 32.76 | 48.21 |
| | GGT | -0.058 | 0.087 | -0.131 | -0.661 | 0.515 | 31.08 | 39.87 |
| Model Summary | | | | | | | | |
| | Model | R | R Square | Adjusted R Square | Std. Error of the Estimate | | | |
| | 1 | 0.449[a] | 0.202 | 0.110 | 8.3065 | | | |

**Table 4. Regression model for liver function tests of obese hypertensive females.**

| Coefficients[a] | | | | | | | | |
|---|---|---|---|---|---|---|---|---|
| Model | | Unstandardized Coefficients | | Standardized Coefficients | T | Sig. | 95.0% Confidence Interval for B | |
| | | B | Std. Error | Beta | | | Lower Bound | Upper Bound |
| 1 | (Constant) | 16.195 | 3.220 | | 5.029 | 0.000 | 9.575 | 22.814 |
| | ALT | -0.091 | 0.187 | -0.154 | -0.484 | 0.632 | -.476 | .294 |
| | AST | 0.139 | 0.156 | 0.308 | 0.889 | 0.382 | -.182 | .460 |
| | GGT | -0.060 | 0.055 | -0.270 | -1.089 | 0.286 | -.173 | .053 |
| Model Summary | | | | | | | | |
| Model | | R | R Square | Adjusted R Square | Std. Error of the Estimate | | | |
| 1 | | 0.235[a] | 0.055 | -0.054 | 9.0379 | | | |

indicators for liver disorder and these elevations are observed mostly due to fatty penetration of the liver [12,15]. Elevated levels of liver enzymes can serve as biomarkers for assessing the severity of CHD especially GGT even in normal limits related to obesity allied hypertension [9,10].

The present study aimed to study the extent of liver enzymes involved in the severity of metabolic syndrome. Among the hepatic enzymes studied, GGT exhibited a strong impact on diabetes and hypertension, thereby, acting as a biomarker for obesity allied ailments. Concentrations of GGT in the liver not only indicate fatty liver volume but also serve as a marker for oxidative stress [16]. Levels of GGT are influenced majorly by obesity and other metabolic ailments like glucose imbalance and hypertension, hence validating the incidence of T2DM and hypertension with elevated GGT levels [17,18]. Hypertension was found to be linked with hepatic enzymes and noticeable elevation in anthropometric measurements, blood pressure, and GGT [19].

Obesity along with polycystic ovary syndrome (PCOS) is the main cause of infertility. In this study, infertile females chosen were obese class I, and obesity with PCOS was the reason for their infertility. In these females, GGT was found to be non-significantly high and linked with PCOS rather than obesity [20].

Aspartate aminotransferase is mainly widespread in the heart along with other parts like the liver, bone, lung, etc. Mitochondrial disturbance due to its oxidative stress or cell death in case of hepatic damage or myocardial infarction leads to enhanced leakage of AST into blood. Hence, its levels can be measured to evaluate the extent of heart and liver damage [21,22]. Negligible elevations in AST levels were noticed in females suffering from diabetes as well as hypertension and the one with diabetes only represented no vital heart and liver damage at this stage of diabetes and its allied issues onset. While it remained within the normal range in hypertension and infertile females, so AST seemed to be unaffected by obesity and have a weak association with it and these findings are consistent with [23].

Abnormal glucose metabolism and NAFLD are mainly responsible for increased levels of ALT apart from hepatitis and these deranged ALT values act as predictors of atherosclerosis and CVD [24,25]. The elevated levels of ALT have been associated with the prevalence of obesity and multiple ailments of MetS such as diabetes mellitus and CVD [26,27].

Outcomes of the present study showed a significant elevation in ALT levels in obese diabetic and obese diabetic hypertensive females, and this is similar to a study that narrated a positive linkage of ALT with increased risk of T2DM incidence [28]. However, a negligible increase was shown by obese infertile and obese diabetic and hypertensive females. High ALT levels expose fatty liver alterations, and these alterations further precede T2DM development

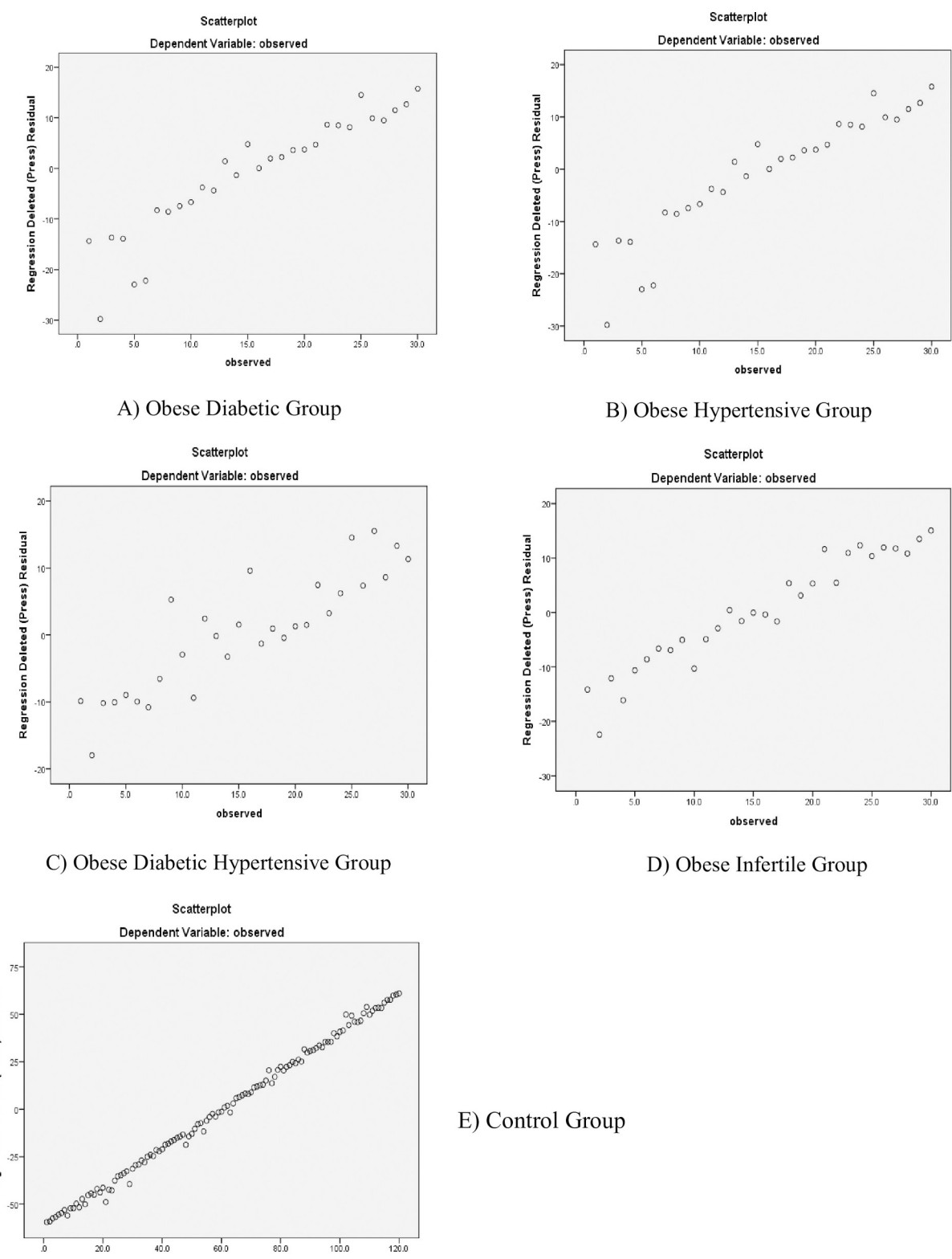

**Fig 3.** A. Scatter plot regression model for liver function tests of obese-induced comorbid and control groups. The figure represents the data after converting all variables into one variable by adding it through SPSS transform.

**Table 5. Regression model for liver function tests of obese diabetic hypertensive females.**

| Coefficients[a] | | | | | | | | |
|---|---|---|---|---|---|---|---|---|
| Model | | Unstandardized Coefficients | | Standardized Coefficients | T | Sig. | 95.0% Confidence Interval for B | |
| | | B | Std. Error | Beta | | | Lower Bound | Upper Bound |
| 1 | (Constant) | 22.824 | 3.963 | | 5.759 | 0.000 | 14.678 | 30.971 |
| | ALT | -0.259 | 0.114 | -0.469 | -2.267 | 0.032 | -.494 | -.024 |
| | AST | 0.002 | 0.077 | 0.005 | 0.026 | 0.979 | -.157 | .161 |
| | GGT | 0.032 | 0.057 | 0.100 | 0.558 | 0.582 | -.085 | .148 |
| Model summary | | | | | | | | |
| Model | | R | R Square | Adjusted R Square | Std. Error of the Estimate | | | |
| 1 | | 0.456[a] | 0.208 | 0.117 | 8.2734 | | | |

[29]. The diabetes epidemic due to GGT and ALT is related to risk factors like BMI and waist-to-hip ratio (WHR) [30]. Thus, elevated ALT levels were found to be associated with hypertension, increased glucose levels, and abdominal obesity [31]. The findings of our study are in line with the findings of previous studies where ALT was found to be associated with general obesity and GGT was associated with both general and abdominal obesity [32]. ALT is also known to have positive associations with fasting blood glucose levels, IR, BMI, and WC along with GGT and these associations are more evident in females than males [23] whereas AST also exhibits positive associations with BMI and WC. The levels of liver enzymes aggravated further with an increase in the various metabolic abnormalities of metabolic syndrome [33]. Due to the presence of outliers in comorbid groups (Fig 3A–3E), the observed results are statistically significant indicating the rejection of the $H_o$ hypothesis which implies that data is not variating equally.

Thus, the study concluded a positive relation between liver enzymes with body weight, especially of ALT and GGT. And since numerous serious health problems such as CVD, HTN, T2DM, and infertility are directly related to overweight and obesity, therefore, enhancement of liver enzymes is predictable in all these morbidities. This study sheds light on the possible role of liver enzymes particularly GGT in the indication of liver pathology accompanied by obesity.

## Conclusions

Obesity paves the way for metabolic syndrome through complications like increased blood pressure, blood sugar levels, cholesterol, and fatty liver. An increase in body weight is notably

**Table 6. Regression model for liver function tests of obese infertile females.**

| Coefficients[a] | | | | | | | | |
|---|---|---|---|---|---|---|---|---|
| Model | | Unstandardized Coefficients | | Standardized Coefficients | T | Sig. | 95.0% Confidence Interval for B | |
| | | B | Std. Error | Beta | | | Lower Bound | Upper Bound |
| 1 | (Constant) | 9.428 | 5.597 | | 1.684 | 0.104 | -2.077 | 20.934 |
| | ALT | 0.059 | 0.225 | 0.055 | 0.261 | 0.796 | -.404 | .521 |
| | AST | 0.196 | 0.212 | 0.199 | 0.924 | 0.364 | -.240 | .632 |
| | GGT | 0.029 | 0.060 | 0.094 | 0.480 | 0.635 | -.095 | .153 |
| Model Summary | | | | | | | | |
| Model | | R | R Square | Adjusted R Square | Std. Error of the Estimate | | | |
| 1 | | 0.233[a] | 0.054 | -0.055 | 9.0417 | | | |

**Table 7. Regression model summary for the control group.**

**Coefficicents**

| Model | | Unstandardized Coefficients | | t | Sig. | 95.0% Confidence Interval for B | |
|---|---|---|---|---|---|---|---|
| | | B | Std. Error | | | Lower Bound | Upper Bound |
| 1 | (Constant) | 59.003 | 9.535 | 6.188 | .000 | 40.119 | 77.888 |
| | ALT (U/L) | .156 | .400 | .390 | .697 | -.637 | .949 |
| | AST (U/L) | -.030 | .348 | -.086 | .931 | -.720 | .660 |
| | GGT (U/L) | -.047 | .283 | -.164 | .870 | -.607 | .514 |
| | **Model Summary** | | | | | | |
| | Model | R | R Square | Adjusted R Square | Std. Error of the Estimate | | |
| | 1 | .916[a] | .808 | .657 | 12.284 | | |

linked with the levels of liver enzymes, especially GGT and ALT. GGT. When the two comorbidities; diabetes and hypertension occurred together they significantly doubled the levels of GGT. Moreover, the liver enzyme alteration was found to be maximum in obese hypertensive and obese infertile females with only 5 and 6% normal production of liver enzymes while the obese diabetic and obese diabetic hypertensive females had 80% of the liver enzymes deranged indicating that obesity along with multiple comorbidities severely impacts the liver functioning and can lead to liver complications and fatty liver diseases. This research concludes that liver enzymes, especially GGT and ALT might serve as biomarkers for identifying the probable presence of metabolic syndrome.

**Table 8. Significance of multiple variables by using ANOVA in regression for liver function tests of obesity-induced metabolic disorders groups.**

| Model | Sum of Squares | Df | Mean Square | F | Sig. |
|---|---|---|---|---|---|
| **Correlation model summary for liver function tests of the obese diabetic group** | | | | | |
| Regression | 1270.731 | 2 | 635.366 | 1.408 | 0.250 |
| Residual | 40606.452 | 90 | 451.183 | | |
| Total | 41877.183 | 92 | | | |
| **Correlation model summary for liver function tests of the obese hypertensive group** | | | | | |
| Regression | 123.705 | 3 | 41.235 | 0.505 | 0.682 |
| Residual | 2123.795 | 26 | 81.684 | | |
| Total | 2247.500 | 29 | | | |
| **Correlation model summary for liver function tests of the obese diabetic hypertensive group** | | | | | |
| Regression | 467.816 | 3 | 155.939 | 2.278 | 0.103 |
| Residual | 1779.684 | 26 | 68.449 | | |
| Total | 2247.500 | 29 | | | |
| **Correlation model summary for liver function tests of the obese infertile group** | | | | | |
| Regression | 121.956 | 3 | 40.652 | 0.497 | 0.687 |
| Residual | 2125.544 | 26 | 81.752 | | |
| Total | 2247.500 | 29 | | | |
| **Correlation model summary for liver function tests of the control group** | | | | | |
| Regression | 372.829 | 3 | 124.276 | 1.724 | 0.0187 |
| Residual | 1874.671 | 26 | 72.103 | | |
| Total | 2247.500 | 29 | | | |

## Supporting information

**S1 File. Consent and questionnaire.**
(PDF)

**S2 File. Inclusion and exclusion criteria.**
(PDF)

**S3 File. Minimal data set.**
(PDF)

**S4 File. Regression analysis equation.**
(PDF)

**S5 File.**
(PDF)

**S6 File.**
(PDF)

**S7 File.**
(PDF)

**S8 File.**
(PDF)

## Acknowledgments

I would like to acknowledge the medical and para-medical staff at Faisal Hospital, Faisalabad for facilitating blood sample collection and laboratory tests.

## Operational definitions

$H_o$ **hypothesis (Null hypothesis):** All the variables show equal variations in all comparable groups.

$H_1$ **hypothesis (Alternate hypothesis):** Each case group show different variations of liver enzymes depending on the status of comorbidities.

## Author Contributions

**Conceptualization:** Tamseela Mumtaz.

**Data curation:** Kainat Tariq.

**Formal analysis:** Tamseela Mumtaz, Kainat Tariq, Khadija Kanwal, Zainab Tariq.

**Investigation:** Kainat Tariq.

**Methodology:** Tamseela Mumtaz.

**Project administration:** Tamseela Mumtaz.

**Software:** Khadija Kanwal, Zainab Tariq.

**Supervision:** Tamseela Mumtaz.

**Validation:** Tamseela Mumtaz, Khadija Kanwal.

**Writing – original draft:** Kainat Tariq.

**Writing – review & editing:** Tamseela Mumtaz.

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
