## [Decision Letter · Decision Letter 0]

7 Nov 2023

PONE-D-23-28668A case control regression analysis of altered liver enzymes in obesity-induced metabolic disordersPLOS ONE

Dear Dr. Mumtaz,

Thank you for submitting your manuscript to PLOS ONE. After careful consideration, we feel that it has merit but does not fully meet PLOS ONE’s publication criteria as it currently stands. Therefore, we invite you to submit a revised version of the manuscript that addresses the points raised during the review process.

Dear author,

Revise the whole manuscript as suggested by reviewers and submit for evaluation.

Thanks

We look forward to receiving your revised manuscript.

Kind regards,

Samiullah Khan, Ph. D

Academic Editor

PLOS ONE

A clean copy of the edited manuscript (uploaded as the new *manuscript* file)”.

Additional Editor Comments:

Dear author,

Revise the whole manuscript carefully as suggested by both reviewers in their comments. Especially rephrase the title of study as advised by reviewer-2. Submit the revised version of manuscript for evaluation.

Thanks

Reviewers' comments:

Reviewer's Responses to Questions

**Comments to the Author**

1. Is the manuscript technically sound, and do the data support the conclusions?

Reviewer #1: Yes

Reviewer #2: Yes

2. Has the statistical analysis been performed appropriately and rigorously? 

Reviewer #1: I Don't Know

Reviewer #2: Yes

3. Have the authors made all data underlying the findings in their manuscript fully available?

Reviewer #1: No

Reviewer #2: Yes

4. Is the manuscript presented in an intelligible fashion and written in standard English?

Reviewer #1: No

Reviewer #2: Yes

5. Review Comments to the Author

Reviewer #1: 1- To control diabetes or blood pressure, patients may have been treated with drugs. How did you separate their effects on liver enzymes from the effects of obesity on liver enzymes?

2- Not consuming alcohol should be an exclusion criteria. 3- Triglyceride, bilirubin and PT-INR - could provide more comprehensive information. Are these data available? 4- In daily practice, ultrasound of the upper abdomen and fibroscan of the liver provide valuable information about liver disease. It is even possible that the liver enzymes are normal and the fibroscan shows moderate degrees of fibrosis. Not using liver imaging is one of the limitations of the study and should be mentioned

Reviewer #2: The authors describe the relationship between obesity and alterations in liver enzymes potentially increasing the risk of liver diseases. The manuscript is well structured with a specific focus on South Asian females. The research sheds light on the unique health dynamics within this demographic.

With that said, I have some suggestions for further improvement:

Title:

The manuscript title is a critical element that sets the tone for the study. To enhance clarity, it may be beneficial to rephrase the title to specify that the study exclusively focuses on data obtained from female individuals. Additionally, consider including the demographic area of the population, for instance, "South Asian Female Population," to provide a more precise context.

Abstract:

I would like to kindly point out that the abstract in the manuscript appears to be incomplete.

Results:

1. Descriptive Titles: I recommend providing more descriptive titles for the ‘Results’ section that directly convey the key observations. This will help readers navigate the content more effectively.

2. Combining WHR and BMI Results: Given the brevity of the WHR and BMI sections and their shared focus on size-related factors, it could be advantageous to combine these into a single section to streamline the presentation.

3. Improving Table Legends: Enhance the clarity of table legends in terms of variable comparisons, particularly in Table 1, and ensure that the context of p-values is clearly defined. If applicable, consider conducting pairwise comparisons to improve data interpretation.

4. Visualization Enhancements: For Figure 1, I suggest utilizing a box plot format with overlaying individual scatter dots, which can provide a clearer representation of the data compared to dynamite plots.

5. Incorporate Regression Dot Plots: In addition to tables, it would be beneficial to present regression dot plots, displaying all data points alongside the main results. Additionally, consider including correlation plots with confidence intervals, and if authors find it suitable, these analyses can be integrated into a single plot.

6. Clear and Accessible Conclusions: Concluding each results section with a clear and less technical statement summarizing the main findings would greatly benefit the manuscript. This will facilitate a more accessible understanding of the research outcomes.

7. Supplementary Equations: To streamline the manuscript, it would be better to move the equations in line number 177/185/194/202/209 (e.g., yˆ=x+y+z) from the main text to a supplementary section, citing them where necessary for clarity.

8. Cholesterol Data and Correlations: To provide a comprehensive context, consider including correlation between cholesterol and obesity. Additionally, clarify whether the authors exclusively measured total cholesterol or if data is available for HDL, LDL, and triglycerides. Correlating this data with obesity would enhance the manuscript's depth.

6. PLOS authors have the option to publish the peer review history of their article (what does this mean?). If published, this will include your full peer review and any attached files.

Reviewer #1: **Yes: **Hafez Fakheri

Reviewer #2: **Yes: **Shrestha Mohapatra

---

## [Author Response · Author response to Decision Letter 0]

7 Jan 2024

All the authors are very grateful for evaluating the manuscript through critical peer review. The reviewers indeed swotted the article with an eagle eye that depicts their competency. They raised such valuable points that greatly altered the study design and its outcome as well. We are much indebted for the reviewers' valuable comments and suggestions that improve the article's quality. The manuscript has been revised in light of reviewer’s recommendation. A detailed, point-to-point response to the reviewers’ comments is as follows

 Reviewer Comments Response

 Reviewer 1 

1 To control diabetes or blood pressure, patients may have been treated with drugs. How did you separate their effects on liver enzymes from the effects of obesity on liver enzymes? It was made sure that patients either did not have any treatment for obesity-related co-morbidities or if treated, they must have shallow potency treatment that could not affect the efficacy of live function. See page 5 lines 107-109 

2 Not consuming alcohol should be an exclusion criterion The study population is assumed to not be addicted to alcohol however to resolve the concern it has been mentioned in the manuscript. See page 5 line 106. 

3 Triglyceride, bilirubin and PT-INR - could provide more comprehensive information. Are these data available Yes, these data are available but are part of another manuscript that is under process, hence cannot be provided. 

4 In daily practice, ultrasound of the upper abdomen and fibro scan of the liver provide valuable information about liver disease. It is even possible that the liver enzymes are normal and the fibro scan shows moderate degrees of fibrosis. Not using liver imaging is one of the limitations of the study and should be mentioned The study was focused on serology and variations in liver enzyme production due to obesity, therefore, a fibro scan or ultrasound of the upper abdomen was not performed. A fibro scan with mild fibrosis may not fulfil the study objectives. This point could be valid if the study focused on liver echotexture rather than its function. However, the authors appreciated the logic, and they conceived the idea that obesity may cause fibrosis which may be asymptomatic with normal liver enzymes 

 Reviewer 2 Response 

1 Title:

The manuscript title is a critical element that sets the tone for the study. To enhance clarity, it may be beneficial to rephrase the title to specify that the study exclusively focuses on data obtained from female individuals. Additionally, consider including the demographic area of the population, for instance, "South Asian Female Population," to provide a more precise context.

 Thanks for bringing this point to our attention. Indeed, it is a very good suggestion. The title has been rephrased as suggested. See title page 1 lines 1-3 

2 Abstract:

I would like to kindly point out that the abstract in the manuscript appears to be incomplete Acknowledged. The word cirrhosis was omitted from the abstract typographically. Issue resolved. see page 2 line 35 

3 Results:

Descriptive Titles: I recommend providing more descriptive titles for the ‘Results’ section that directly convey the key observations. This will help readers navigate the content more effectively. Titles of results sections have been changed that indicate the key observations of results being discussed.

4 Combining WHR and BMI Results: Given the brevity of the WHR and BMI sections and their shared focus on size-related factors, it could be advantageous to combine these into a single section to streamline the presentation. The sections have been combined. See title page 6 lines 129-130

5 Improving Table Legends: Enhance the clarity of table legends in terms of variable comparisons, particularly in Table 1, and ensure that the context of p-values is clearly defined. If applicable, consider conducting pairwise comparisons to improve data interpretation. Table 1 legends are described more clearly indicating P=0.05 and pairwise comparisons are indicated in the form of percentage increase or decrease. Table 2 Page # 8

 Visualization Enhancements: For Figure 1, I suggest utilizing a box plot format with overlaying individual scatter dots, which can provide a clearer representation of the data compared to dynamite plots. Thank you for your valuable suggestion scatter plot provides a better illustration of data. Figure 1 is updated with individual scatter dots. See Fig 1 

 Incorporate Regression Dot Plots: In addition to tables, it would be beneficial to present regression dot plots, displaying all data points alongside the main results. Additionally, consider including correlation plots with confidence intervals, and if authors find it suitable, these analyses can be integrated into a single plot. Regression dot plots incorporated as suggested. See Fig. 7. 

The correlation plots with confidence intervals cannot be integrated into a single plot, thus presented in a table format. See page no.15 Table 8 

 Clear and Accessible Conclusions: Concluding each results section with a clear and less technical statement summarizing the main findings would greatly benefit the manuscript. This will facilitate a more accessible understanding of the research outcomes. Done as suggested.

 Supplementary Equations: To streamline the manuscript, it would be better to move the equations in line number 177/185/194/202/209 (e.g., yˆ=x+y+z) from the main text to a supplementary section, citing them where necessary for clarity. The equations have been moved from the main file to the supplementary file (S8 File).

 Cholesterol Data and Correlations: To provide a comprehensive context, consider including the correlation between cholesterol and obesity. Additionally, clarify whether the authors exclusively measured total cholesterol or if data is available for HDL, LDL, and triglycerides. Correlating this data with obesity would enhance the manuscript's depth. We appreciate your suggestion; indeed, it has given a new dimension to our manuscript as we have to go through the results once again thoroughly. Correlation between cholesterol and BMI is done as suggested in each respective comorbid group. See Fig. 2-6. Further data about HDL, LDL, and triglycerides cannot be provided now because it is used in another manuscript that is under process.

Hope the answer will satisfy the reviewers and make the understanding of the research better. We believe that manuscript is now suitable for publication in PLOS ONE

---

## [Decision Letter · Decision Letter 1]

8 Feb 2024

PONE-D-23-28668R1A case-control regression analysis of liver enzymes in obesity-induced metabolic disorders in South Asian femalesPLOS ONE

Dear Dr. Mumtaz,

Thank you for submitting your manuscript to PLOS ONE. After careful consideration, we feel that it has merit but does not fully meet PLOS ONE’s publication criteria as it currently stands. Therefore, we invite you to submit a revised version of the manuscript that addresses the points raised during the review process.

We look forward to receiving your revised manuscript.

Kind regards,

Samiullah Khan, Ph. D

Academic Editor

PLOS ONE

Journal Requirements:

**Additional Editor Comments:**

Dear author,

Revise the manuscript by incorporating the all corrections suggested by reviewer#2 and submit the re-revised manuscript.

Reviewers' comments:

Reviewer's Responses to Questions

**Comments to the Author**

1. If the authors have adequately addressed your comments raised in a previous round of review and you feel that this manuscript is now acceptable for publication, you may indicate that here to bypass the “Comments to the Author” section, enter your conflict of interest statement in the “Confidential to Editor” section, and submit your "Accept" recommendation.

Reviewer #1: (No Response)

Reviewer #2: All comments have been addressed

2. Is the manuscript technically sound, and do the data support the conclusions?

Reviewer #1: (No Response)

Reviewer #2: Partly

3. Has the statistical analysis been performed appropriately and rigorously? 

Reviewer #1: I Don't Know

Reviewer #2: No

4. Have the authors made all data underlying the findings in their manuscript fully available?

Reviewer #1: Yes

Reviewer #2: Yes

5. Is the manuscript presented in an intelligible fashion and written in standard English?

Reviewer #1: Yes

Reviewer #2: Yes

6. Review Comments to the Author

Reviewer #1: (No Response)

Reviewer #2: Dear Authors,

1. Please mention the post-hoc tests performed after ANNOVA in the methods and the figure legends.

2. Please mention the number of samples i.e., n=? at appropriate places including figure legends.

3. Please write the key observation as the title of each result instead of writing the method and the statistical test used. For example, result 1 shows the relationship of obesity, diabetes, and hypertension on key health indicators in women.

4. In Figure 2,3,4,5 & 6:

1. Please include Correlation coefficient (R2) and the P-value on the regression plots.

2. Please mention axis titles.

3. It would be visually appealing to put all these plots in a single page.

4. All the plots should have consistent Y axis limits for enhanced clarity and interpretability.

Thank you.

7. PLOS authors have the option to publish the peer review history of their article (what does this mean?). If published, this will include your full peer review and any attached files.

Reviewer #1: No

Reviewer #2: **Yes: **Shrestha Mohapatra

---

## [Author Response · Author response to Decision Letter 1]

23 Apr 2024

The authors are very much obliged for the suggestion of the reviewer 2 for valuable comments. The suggestion to express the results as a title rather than methods or statistical analysis changed the whole impact of the article. We have tried to address all the points raised by the reviewers except one that the Y-axis should have consistent limits. We are unable to make the Y-axis values consistent because every variable has a different unit value and can’t be set at uniformity. All regression dot plots are also incorporated on a single page to explain the results more obviously.

---

## [Editor Report · Decision Letter 2]

2 May 2024

A case-control regression analysis of liver enzymes in obesity-induced metabolic disorders in South Asian females

PONE-D-23-28668R2

Dear Dr. Tamseela,

We’re pleased to inform you that your manuscript has been judged scientifically suitable for publication and will be formally accepted for publication once it meets all outstanding technical requirements.

Kind regards,

Samiullah Khan, Ph. D

Academic Editor

PLOS ONE

Additional Editor Comments (optional):

Dear author, the manuscript is now suitble for publication in PLoS ONE because all the corections have been done and addressed all the queries raised by the reviewers.
---

## [Editor Report · Acceptance letter]

14 May 2024

PONE-D-23-28668R2 

PLOS ONE

Dear Dr. Mumtaz, 

I'm pleased to inform you that your manuscript has been deemed suitable for publication in PLOS ONE. Congratulations! Your manuscript is now being handed over to our production team.

Kind regards, 

on behalf of

Dr. Samiullah Khan 

Academic Editor

PLOS ONE